# Risk of colorectal cancer in patients with alcoholism: A nationwide, population-based nested case-control study

Tzu-Chiao Lin[1,2], Wu-Chien Chien[3,4,5], Je-Ming Hu[1,2,6], Nian-Sheng Tzeng[2,7,8], Chi-Hsiang Chung[3,4,9], Ta-Wei Pu[1,2], Cheng-Wen Hsiao[1,2], Chao-Yang Chen[1,2]*

1 Division of Colorectal Surgery, Department of Surgery, Tri-Service General Hospital, National Defense Medical Center, Taipei, Taiwan, Republic of China, 2 School of Medicine, National Defense Medical Center, Taipei, Taiwan, Republic of China, 3 School of Public Health, National Defense Medical Center, Taipei, Taiwan, Republic of China, 4 Department of Medical Research, Tri-Service General Hospital, National Defense Medical Center, Taipei, Taiwan, Republic of China, 5 Graduate Institute of Life Sciences, National Defense Medical Center, Taipei, Taiwan, Republic of China, 6 Graduate Institute of Medical Sciences, National Defense Medical Center, Taipei, Taiwan, Republic of China, 7 Department of Psychiatry, Tri-Service General Hospital, School of Medicine, National Defense Medical Center, Taipei, Taiwan, Republic of China, 8 Student Counseling Center, National Defense Medical Center, Taipei, Taiwan, Republic of China, 9 Taiwanese Injury Prevention and Safety Promotion Association, Taipei, Taiwan, Republic of China

* cartilage77@yahoo.com.tw

**Data Availability Statement:** All relevant data are within the paper and its Supporting Information files.

## Abstract

### Background

Colorectal cancer (CRC) is regarded as a multifactorial disease and shares many risk factors with alcoholism. However, the association between alcoholism and CRC remains controversial.

### Objectives

In this study, we aimed to evaluate the association between alcoholism and risk of CRC.

### Methods

We performed a large-scale, population-based nested case-control study using the Longitudinal Health Insurance Database 2013, derived from Taiwan's National Health Insurance Research Database, and collected data from 2000 to 2013. There were 49,095 diagnosed cases of CRC defined according to the International Classification of Diseases, Ninth Revision, Clinical Modification. Each case was matched with three controls by sex, age, index date of CRC, and annual medical visits; a total of 147,285 controls were identified. Multiple risk factors of CRC in alcoholism cases were investigated using unconditional multiple logistic regression analysis.

### Results

Among 49,095 cases of CRC, alcoholism was associated with a significantly higher risk of CRC (adjusted odds ratio (OR), 1.631; 95% CI, 1.565–1.699) in multivariate logistic regression, after adjusting other CRC risk factors, and in stratified analysis with multivariate logistic

**Funding:** This study was supported by grants from Tri-Service Hospital Research Foundation (TSGH-B-109010). No additional external funding was received for this study. All of the authors received no specific funding for this work. The funders had no role in study design, data collection and analysis, decision to publish, or preparation of the manuscript.

**Competing interests:** All of the authors declared that no competing interests exist.

regression. In addition, there was a time-dependent relationship between alcoholism duration and CRC risk in >1 year, > 2 years, >5 years, and > 11 years groups (adjusted ORs, 1.875, 2.050, 2.662 and 2.670; 95% CI, 1.788–1.967, 1.948–2.158, 2.498–2.835, and 2.511–2.989 respectively).

## Conclusion

An association between alcoholism and risk of CRC was found in this study. Furthermore, patients with longer alcoholism history showed higher likelihood of developing CRC, which indicates a time-dependent relationship between alcoholism exposure and CRC. Further research on colorectal tumorigenesis is needed.

## Introduction

Alcohol use disorder (AUD) is characterized by habitual alcohol consumption, loss of control over alcohol intake, having a negative emotional state when not using alcohol, and is considered a composite of alcohol dependence syndrome and alcoholism[1, 2]. The 2016 World Health Organization Global Burden of Diseases, Injuries, and Risk Factors Study warned that alcohol use was the seventh leading cause of death and the highest factor for disability-adjusted life-years worldwide[3, 4]. Alcohol consumption seems to have a strong relationship with developing cancers of the oral cavity, pharynx, esophagus, stomach, colorectum, central nervous system, pancreas, breast, and prostate[5, 6]. Colorectal cancer (CRC) is a major health challenge and has become the most common cancer in Taiwan, with the third leading age-standardized mortality rate among all fatal cancers[7, 8].

Several studies have reported that CRC is a multifactorial disease process and seems to share many risk factors with alcoholism[9, 10], which might have direct effects on colon carcinogenesis processes via multiple molecular metabolites, including oxidative and non-oxidative byproducts and changes in gut environmental factors that can progressively affect genetic alterations, followed by cancer cell proliferation[11]. However, the association between alcohol and CRC remains controversial. Some studies have revealed alcohol consumption as an independent and important risk factor for CRC that may also influence prognosis and enhance mortality rate[12, 13]. Other studies have showed no or even an inverse relationship between alcohol and CRC, particularly wine[14–16].

To extend the understanding between risk of CRC and alcoholism, we conducted a large, nationwide, population-based nested case-control study using the Taiwan's National Health Insurance Research Database (NHIRD) to investigate the effect of alcoholism on CRC risk.

## Methods

### Data source

The National Health Insurance program (NHIP), run by the government of Taiwan, was started in 1995 and covers more than 99% of the Taiwanese population (i.e., more than 23 million beneficiaries per year). The NHIRD was derived in 1997 from the NHIP, which contains all original claims data and medical records including the characteristics of ambulatory care, hospital inpatient care, patients seen in emergency departments, prescription records, and disease diagnoses. Therefore, the database is representative of the population of Taiwan and had been used for health economics and research on health care utilization, preventive healthcare, and other medical research purposes, in addition to national health policy[17]. We analyzed

the Longitudinal Health Insurance Database (LHID), which is a dataset of one million people randomly sampled in 2013 from the NHIRD, with identifying information of each patient encrypted to protect privacy. Among all beneficiaries, we collected data on 989,753 individuals with 26,769,418 medical events from January 1, 2000, to December 31, 2013. The conformation and features of individuals registered in the LHID were normally distributed. Owing to the ability to easily link with data of population-based screening for cancers using personal patient identification numbers, the selected sample of patients with CRC was free of selection and participation bias[17, 18].

In Taiwan, strict measures are adopted for cancer diagnoses. Insured patients with CRC are diagnosed by a clinical physician with confirmation through the pathological examination of tissue biopsy, and the inclusion criteria of alcoholism and CRC are well defined on the basis of International Classification of Diseases, Ninth Revision, Clinical Modification (ICD-9-CM). To register the details of diagnosed patients in their catastrophic illness certificates, coding of diagnoses are processed rigorously by clinical physicians, with formal reviews conducted by medical professionals of the National Health Insurance Administration (NHIA)[19]. Therefore, these data are monitored and strictly evaluated by the Bureau of National Health Insurance for reimbursement purposes, and the diagnoses of alcoholism and CRC are highly reliable.

## Study design, participants, and ethics

We conducted a nested case-control study to examine the association between alcoholism and CRC. We used ICD-9-CM coding system to identify diagnoses. CRC is defined as ICD-9-CM codes 153, 153.0–153.3, 153.6–153.9, 154, 154.0–154.3, 154.8, and 159. Alcoholism is defined as ICD-9-CM codes 291.0–291.5, 291.81, 291.89, 291.9, 303.00–303.03, 303.90–303.93, 350.00–350.03, 571.0–571.3, 571.40, 571.41, 571.49, V11.3, V61.41, and V79.1. Other types of malignancy are defined as ICD-9-CM codes 140–208.

We enrolled patients with at least three outpatient or inpatient visits who were diagnosed between 2000 and 2013, to ensure validity. We excluded beneficiaries with any prior history of cancer diagnosed before CRC, patients younger than 18 years old, and those with unknown or missing information of sex. Patients without CRC in the control group were randomly selected from the LHID using a strategy for matching according to sex, age (with the same birth year), the index date (date of the first diagnosed record of CRC), and annual medical visits. The ratio for selection was 1:3 (case:control). The same exclusion criteria were applied to the control group. Over the 14-year study period, we identified 49,095 cases of CRC and acquired 147,285 patients as a control group (Fig 1).

This study was approved by the Institutional Review Board of Tri-Service General Hospital (TSGHIRB NO. 2-106-05-029).

## Variables of interest, comorbidities

Sociodemographic characteristics are recorded in the LHID, including age, sex, insurance premium, location, hospital level, residential urbanization level, and other comorbidities. The 368 cities/towns in Taiwan are divided into four groups ranked by urbanization level according to several indicators and based on population size. Level 1 refers to the most urbanized areas and Level 4 refers to the least urbanized areas based on an NHIA report. Comorbid diseases were identified as follows: chronic obstructive pulmonary disease (COPD) (ICD-9-CM codes 490–496), diabetes mellitus (DM) (ICD-9-CM code 250), coronary artery disease (CAD) (ICD-9-CM codes 410–414), hypertension (HTN) (ICD-9-CM codes 401–405), hypercholesterolemia (ICD-9-CM codes 272.0–272.4), peptic ulcer (ICD-9-CM codes 531–533), inflammatory bowel disease (IBD) (ICD-9-CM codes 555–556), and colorectal polyp (ICD-9-CM codes

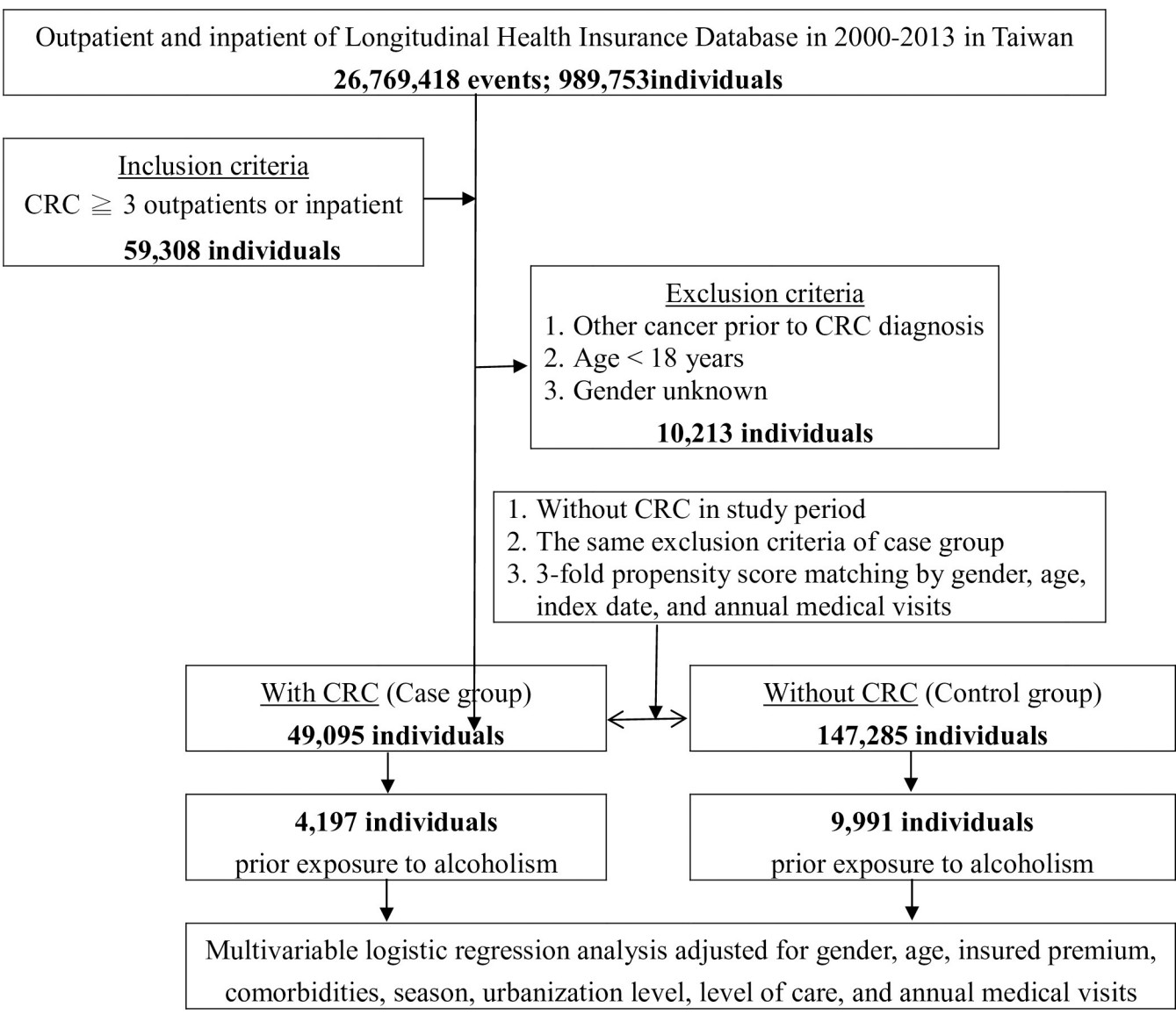

**Fig 1. The flowchart of study design (nested case-control study) from National Health Insurance Research Database in Taiwan.**

211.3–211.4, 569.0). Epidemiologic studies have consistently reported a positive association between obesity, smoking and CRC[20, 21]. Nevertheless, data on obesity and cigarette smoking were not available in the NHIRD. In the present study, obesity-related diseases (ICD-9-CM code 250, DM; ICD-9-CM codes 401–405, HTN; ICD-9-CM codes 272.0–272.2, and 272.4, hypercholesterolemia; and ICD-9-CM codes 410–414, CAD) were used as surrogates of obesity, and smoking-related diseases (ICD-9-CM codes 490–496, COPD) were used as surrogates of cigarette smoking; these were treated as potential confounders in multivariable logistic regression analysis.

## Statistical analysis

Distributions of sociodemographic data and comorbidities were compared between CRC cases and controls using the chi-squared test for categorial variables and *t*-test for continuous

variables. Unconditional multiple logistic regression analyses were performed to evaluate the risks of CRC associated with alcoholism, adjusted for the confounding effects of age, sex, insurance premium level, and all abovementioned comorbidities. Adjusted models with significant covariates were constructed using backward selection with the likelihood ratio test. Risk factors of CRC analyses were conducted with stratification by different variables. Further analysis was performed for assessing the dose-response effect of alcoholism on risk of CRC according to the duration in years of alcoholism. SAS 9.4 software (SAS Institute, Cary, NC, USA) was used for data analysis. A significance level of $P < .05$ was used in two-sided tests.

## Results

Data were recruited from the LHID from 2000 to 2013, including data of a total 989,753 individuals database in Taiwan. There were 59,308 patients included in this analysis who had at least three outpatient or inpatient visits with a CRC diagnosis. After applying the exclusion criteria, 49,095 patients were identified for inclusion in the case group. For the control group, we applied the same exclusion criteria and used a threefold propensity score matching strategy by sex, age, index date, and annual medical visits, thus identifying 147,285 patients for inclusion in the control group.

Table 1 lists various demographic characteristics of the case and control groups and highlights significant differences between the two groups. The findings revealed that alcoholism, patients' insurance premium, comorbidities, location, residential urbanization level, and hospital level of care were significantly different between the case and control groups. In particular, patients in the case group were more likely to have comorbidities, including COPD, DM, CAD, HTN, hypercholesterolemia, peptic ulcer, IBD, and colorectal polyp, than the control group (P < .05). In most patients with CRC, their disease was diagnosed and treated in Northern, Middle and Southern Taiwan, which have a combination of urbanization level 1 and 2 cities, and these patients were predominantly treated in medical centers.

In Table 2, the differences between the case and control group were analyzed in multivariate logistic regression. After adjusting other CRC risk factors of age, insurance premium, comorbidities, location, and urbanization level, the alcoholism group had a consistently higher risk of CRC than the control group (adjusted odds ratio (OR), 1.631; 95% CI, 1.565–1.699). In Table 3, we conducted a stratified analysis with multivariable logistic regression for the factors between case and control groups. The results indicated that participants with previous alcoholism had higher CRC risk, regardless of sex (adjusted ORs, 3.103 and 1.291; 95% CIs, 2.870–3.358 and 1.224–1.358, respectively, in male and female participants). For age groups 45–64 years, and 65 years or older, participants with alcoholism also showed higher CRC risk (adjusted ORs, 1.110 and 2.649; 95% CIs, 1.028–1.174 and 2.497–2.811, respectively). When we applied stratified analysis according to insurance premium, alcoholism history was associated with increased risk of CRC in both low (premium < 18,000 NTD) and medium (premium 18,000~34,999 NTD) level categories: adjusted ORs (95% CIs) were 1.510 (1.081–2.120) and 1.633 (1.565–1.702), respectively. For comorbidities such as COPD, DM, CAD, HTN, hypercholesterolemia, peptic ulcer, IBD, and colorectal polyp, similar associations were found in that previous alcoholism presented higher CRC risk. Furthermore, when we stratified participants according to residential urbanization levels (the highest level 1 to the lowest level 4) and healthcare levels (hospital center, regional hospital and local hospital), the data revealed that patients with prior alcoholism had significantly greater CRC risk across all patient visit variables (urbanization levels adjusted ORs, 1.790, 1.690, 1.668, and 1.335; 95% CIs, 1.657–1.934, 1.589–1.798, 1.436–1.940, and 1.211–1.470 respectively; healthcare levels adjusted ORs, 1.987, 1.515, and 1.456; 95% CIs, 1.645–2.412, 1.427–1.599, and 1.401–1.708 respectively).

**Table 1. Demographic characteristics and comorbidities in the study with and without colorectal cancer.**

| | CRC | | |
| --- | --- | --- | --- |
| | **With** | **Without** | |
| **Variables** | **N = 49095** | **N = 147285** | *P value* |
| **Alcoholism** | **N (%)** | **N (%)** | <0.001 |
| Without | 44,898(91.45) | 137,294(93.22) | |
| With | 4,197(8.55) | 9,991(6.78) | |
| **Gender** | | | 0.999 |
| Male | 28,325(57.69) | 84,975(57.69) | |
| Female | 20,770(42.31) | 62,310(42.31) | |
| **Age group (years)** | | | 0.999 |
| 18–44 | 3,593(7.32) | 10,779(7.32) | |
| 45–64 | 17,649(35.95) | 52,947(35.95) | |
| ≧65 | 27,853(56.73) | 83,559(56.73) | |
| **Insured premium (NT$)** | | | <0.001 |
| <18,000 | 46,235(94.17) | 136,902(92.95) | |
| 18,000–34,999 | 1,897(3.86) | 7,820(5.31) | |
| ≧35,000 | 963(1.96) | 2,563(1.74) | |
| **Comorbidities** | | | |
| **COPD** | 6,424(13.08) | 17,404(11.82) | <0.001 |
| **DM** | 12,764(26.00) | 27,644(18.77) | <0.001 |
| **CAD** | 9,177(18.69) | 24,623(16.72) | <0.001 |
| **HTN** | 20,709(42.18) | 42,874(29.11) | <0.001 |
| **Hypercholesterolemia** | 3,614(7.36) | 13,204(8.96) | <0.001 |
| **Peptic ulcer** | 10,192(20.76) | 18,140(12.32) | <0.001 |
| **IBD** | 212(0.43) | 514(0.35) | 0.009 |
| **Colorectal polyp** | 740(1.51) | 1006(0.68) | <0.001 |
| **Location** | | | <0.001 |
| Northern Taiwan | 19,906(40.55) | 53,168(36.10) | |
| Middle Taiwan | 13,361(27.21) | 44,371(30.13) | |
| Southern Taiwan | 13,305(27.10) | 37,306(25.33) | |
| Eastern Taiwan | 2,495(5.08) | 11,544(7.84) | |
| Outlets islands | 28(0.06) | 896(0.61) | |
| **Urbanization level** | | | <0.001 |
| 1 (The highest) | 17,466(35.58) | 38,410(26.08) | |
| 2 | 21,993(44.80) | 64,267(43.63) | |
| 3 | 2,718(5.54) | 12,923(8.77) | |
| 4 (The lowest) | 6,918(14.09) | 31,685(21.51) | |
| **Level of care** | | | <0.001 |
| Hospital center | 20,700(42.16) | 38,393(26.07) | |
| Regional hospital | 22,555(45.94) | 72,614(49.30) | |
| Local hospital | 5,840(11.90) | 36,278(24.63) | |
| **Annual medical visits** | 9.82±10.34 | 9.75±10.71 | 0.206 |

CRC = colorectal cancer, COPD = chronic obstructive pulmonary disease, DM = diabetes mellitus, CAD = coronary artery disease, HTN = hypertension,

IBD = inflammatory bowel disease, *P*: Chi-square / Fisher exact test on category variables and t-test on continue variables

In subsequent analysis, we additionally stratified participants according to duration of alcoholism in multivariable logistic regression, to further examine the association between alcoholism and CRC. The duration of alcoholism was divided into four groups (> 1 year, > 2 years, >

**Table 2. Factors of CRC by using multivariable logistic regression.**

| Variables | Adjusted OR | 95% CI |
|---|---|---|
| **Alcoholism** | | |
| Without | 1 (Reference) | |
| With | 1.631 | (1.565–1.699) *** |
| **Gender** | | |
| Male | 1.019 | (0.997–1.042) |
| Female | 1 (Reference) | |
| **Age group (years)** | | |
| 18–44 | 1 (Reference) | |
| 45–64 | 1.038 | (0.993–1.086) |
| ≧65 | 1.242 | (1.187–1.300) *** |
| **COPD** | | |
| Without | 1 (Reference) | |
| With | 1.644 | (1.625–1.665) *** |
| **DM** | | |
| Without | 1 (Reference) | |
| With | 1.814 | (1.793–1.835) *** |
| **CAD** | | |
| Without | 1 (Reference) | |
| With | 1.666 | (1.647–1.685) *** |
| **HTN** | | |
| Without | 1 (Reference) | |
| With | 1.724 | (1.706–1.742) *** |
| **Hypercholesterolemia** | | |
| Without | 1 (Reference) | |
| With | 0.500 | (0.480–0.520) *** |
| **Peptic ulcer** | | |
| Without | 1 (Reference) | |
| With | 1.007 | (0.961–1.035) |
| **IBD** | | |
| Without | 1 (Reference) | |
| With | 1.479 | (1.247–1.755) *** |
| **Colorectal polyp** | | |
| Without | 1 (Reference) | |
| With | 2.195 | (1.987–2.432) *** |

OR = odds ratio, CI = confidence interval, Adjusted OR = Adjusted odds ratio: Adjusted for the variables listed in Table 1.

$^{*}P < .05$, $^{**}P < .01$,

$^{***}P < .001$

5 years, and > 11 years); the final results are presented in Table 4. The results of analysis showed a possible time-dependent relationship between alcoholism duration and CRC risk in the > 1 year, > 2 years, > 5 years, and > 11 years groups (adjusted ORs, 1.875, 2.050, 2.662 and 2.670; 95% CI, 1.788–1.967, 1.948–2.158, 2.498–2.835, and 2.511–2.989 respectively). In other words, patients who had a longer alcoholism history showed higher likelihood of CRC.

**Table 3. Factors of CRC stratified by variables listed in the table by using multivariable logistic regression.**

|  | With Alcoholism *vs.* without Alcoholism | |
| --- | --- | --- |
| **Stratified** | **Ratio** | **Adjusted OR (95%CI)** |
| **Total** | 1.260 | 1.631(1.565–1.699)*** |
| **Gender** |  |  |
| Male | 1.284 | 3.103(2.870–3.358)*** |
| Female | 1.214 | 1.291(1.224–1.358)*** |
| **Age group (years)** |  |  |
| 18–44 | 1.021 | 1.078(0.914–1.255) |
| 45–64 | 1.068 | 1.110(1.028–1.174)** |
| ≧65 | 1.501 | 2.649 (2.497–2.811)*** |
| **COPD** | 1.311 | 1.808(1.650–1.980)*** |
| **DM** | 1.254 | 1.688(1.604–1.775) *** |
| **CAD** | 1.388 | 1.771(1.632–1.922) *** |
| **HTN** | 1.350 | 1.751(1.651–1.858) *** |
| **Hypercholesterolemia** | 1.140 | 1.414(1.256–1.592) *** |
| **Peptic ulcer** | 1.176 | 2.101(1.993–2.211) *** |
| **IBD** | 1.818 | 1.757(1.022–3.100) ** |
| **Colorectal polyp** | 1.286 | 1.879(1.592–2.304) *** |
| **Urbanization level** |  |  |
| 1 (The highest) | 1.518 | 1.790(1.657–1.934) *** |
| 2 | 1.349 | 1.690(1.589–1.798) *** |
| 3 | 1.276 | 1.668(1.436–1.940) *** |
| 4 (The lowest) | 1.040 | 1.335(1.211–1.470) *** |
| **Level of healthcare** |  |  |
| Hospital center | 1.729 | 1.987(1.645–2.412) *** |
| Regional hospital | 1.252 | 1.515(1.427–1.599) *** |
| Local hospital | 1.137 | 1.456(1.401–1.708) *** |

Ratio = With/Without alcoholism; Adjusted OR = Adjusted odds ratio: Adjusted for the variables listed in Table 2.;
CI = confidence interval

*$P < .05$, **$P < .01$,

***$P < .001$

## Discussion

This large-scale, population-based study showed that alcoholism was an independent risk factor for CRC (adjusted OR, 1.631; 95% CI, 1.565–1.699). To the best of our knowledge, this study is the first large population-based analysis that shows a possible direct correlation between CRC and alcoholism. In addition, patients who had longer alcoholism exposure history showed higher likelihood of developing CRC. In other studies, alcohol consumption can be only estimated by means of questionnaires to evaluate the effects of dose-response relationship among heavy drinkers with high risks for developing CRC. However, our study illustrated a time-dependent correlation between alcoholism exposure duration and increased CRC, which is demonstrated by a 17 years follow-up database[14–16, 22].

The topic of alcoholism and CRC sparked much interest among researchers. The development of CRC is recognized as a multifactorial disease process[10], which is typically considered to be affected by hereditary components, genetic factors, diet, lifestyle, environmental factors (including obesity, sedentary behavior, smoking, and alcohol consumption, among others), inflammatory conditions of the digestive tract (encompassing ulcerative colitis and

**Table 4. CRC among different retrospective duration by using multivariable logistic regression.**

| Retrospective duration | With CRC (N = 49,095) | Without CRC (N = 147,285) | With CRC vs. Without CRC |
|---|---|---|---|
| | Alcoholism exposure N (%) | | Adjusted OR (95%CI) |
| Overall | 4,197 (8.55) | 9,991 (6.78) | 1.631 (1.565–1.699) *** |
| > 1 year | 2,980 (6.07) | 7,884 (5.35) | 1.875 (1.788–1.967) *** |
| > 2 years | 2,513 (5.12) | 6,681 (4.54) | 2.050 (1.948–2.158) *** |
| > 5 years | 1,552 (3.16) | 4,260 (2.89) | 2.662 (2.498–2.835) *** |
| > 11 years | 614 (1.25) | 1,596 (1.08) | 2.670 (2.511–2.989) *** |

Adjusted OR = Adjusted odds ratio: Adjusted for the variables listed in Table 2.; CI = confidence interval

Retrospective duration: The last alcoholism exposure diagnosis before the first CRC diagnosis

>11 years: 1997–2013

*$P < .05$, **$P < .01$,

***$P < .001$

Crohn disease), and acquired somatic mutations with normal colonic mucosa dysplasia that finally lead to pre-neoplastic polyps. An epidemiological study demonstrated that long-term alcohol consumption leads to an increased mortality rate and incidence for cancer[23]. This is proven by a recent meta-analysis that utilize questionnaires to estimate alcohol consumption, which have indicated a J-shaped dose-response relationship between alcohol consumption and CRC mortality, with an overall increased risk for heavy drinkers[24, 25]. In conclusion, our study of NHIRD dataset with 17 years follow-up emphasized the direct correlation between alcoholism exposure history and CRC using the time-dependent relationship method.

Previous studies have reported that alcohol consumption has several direct and indirect effects that contribute to carcinogenesis. For instance, acetaldehyde (the first metabolite of ethanol oxidation) seems to be the most toxic metabolite owing to its mutagenic and carcinogenic properties. Moreover, alcohol and its metabolites may induce cancer-promoting cascades, such as DNA adduct formation, oxidative stress, lipid peroxidation, genetic and epigenetic alterations, intestinal epithelial barrier dysfunction, and immune modulatory effects[5, 26, 27]. Besides, alcohol has been thought to be linked with increased risk of some specific type of colorectal polyps by the accumulation of pathogenic mutations and finally lead to CRC[28, 29]. Another study also postulated that the ADH1C polymorphism and ALDH2*2 allele have strong relationships with CRC[30]. Poor dietary habits, low levels of folate (including decreased folate absorption, and inhibition of folate cycle enzymes) and fiber consumption, as well as circadian rhythm disruption, are thought to directly lead to alcohol-induced colon carcinogenesis[31]. Furthermore, chronic alcohol consumption may also cause intestinal inflammation, which includes changes in intestinal microbiota composition and distribution, increased absorbency of the intestinal epithelium, and alteration of intestinal immune equilibrium[11, 32–34].

Notwithstanding the strengths of our study, the present study had several limitations. First, the diagnoses of alcoholism and CRC were based on diagnostic codes registered by a physician in the claims database of the NHIP; therefore, some registration biases might be involved in the CRC risk, acting as potential confounding factors, which were not excluded from the study. Second, patients with asymptomatic alcoholism may not seek health care until they experience considerable discomfort, leading to underestimation of the CRC risk among patients with alcoholism. Third, information on important confounders for the association between alcoholism and CRC risk, such as family history, obesity, smoking habits, and dietary patterns, were unavailable in the NHIRD. In the present study, we used the diagnostic codes

for COPD as a proxy for the confounding covariate of smoking because COPD is well known to be related to smoking habits. In addition, obesity is closely related to DM, HTN, hyperlipidemia, and CAD. The aforementioned factors were adjusted for in partial exclusion of the confounding effect of obesity. However, we were unable to control for the confounding effect of dietary patterns. In the literature, a meta-analysis revealed a significant positive association between a Westernized diet and the risk of CRC[35].

Despite these limitations, the use of a nationwide population database, which provided a large sample size, and adjustment for certain covariates related to alcoholism enabled us to detect a significant relation between alcoholism and the risk of developing CRC. Moreover, we explored the possible time-dependent effect of alcoholism on CRC risk. We further found that the longer the alcoholism history; the higher the likelihood of CRC risk. The onset of alcoholism occurs relatively early in life; therefore, early identification of lifestyle risk factors and symptoms are critical to preventing the development of CRC. Besides, the data in our study showed that alcoholism exposure had no significant impact on the survival of CRC patients (survival rate of 90.04% vs. 90.82% respectively, in alcohol exposure and non-exposure groups, $P$ = 0.095), and the CRC patients with alcoholism exposure had higher medical cost than those without alcoholism exposure (S1 & S2 Tables). Further randomized large-scale prospective studies are needed to confirm these issues, and the effects of management of alcoholism on colorectal tumorigenesis should be elucidated.

## Supporting information

**S1 Table. Survival among CRC patients with and without alcoholism exposure.**
(DOCX)

**S2 Table. Medical cost of the CRC with and without alcoholism exposure.**
(DOCX)

## Author Contributions

**Conceptualization:** Tzu-Chiao Lin, Je-Ming Hu, Ta-Wei Pu, Cheng-Wen Hsiao, Chao-Yang Chen.

**Data curation:** Tzu-Chiao Lin, Wu-Chien Chien, Je-Ming Hu, Nian-Sheng Tzeng, Chi-Hsiang Chung, Ta-Wei Pu, Chao-Yang Chen.

**Formal analysis:** Tzu-Chiao Lin, Chao-Yang Chen.

**Investigation:** Tzu-Chiao Lin, Wu-Chien Chien, Nian-Sheng Tzeng, Cheng-Wen Hsiao, Chao-Yang Chen.

**Methodology:** Tzu-Chiao Lin, Wu-Chien Chien, Nian-Sheng Tzeng, Chi-Hsiang Chung, Ta-Wei Pu, Cheng-Wen Hsiao, Chao-Yang Chen.

**Resources:** Tzu-Chiao Lin, Je-Ming Hu, Chi-Hsiang Chung, Ta-Wei Pu, Chao-Yang Chen.

**Software:** Chi-Hsiang Chung.

**Supervision:** Tzu-Chiao Lin, Je-Ming Hu, Cheng-Wen Hsiao, Chao-Yang Chen.

**Validation:** Tzu-Chiao Lin.

**Visualization:** Tzu-Chiao Lin.

**Writing – original draft:** Tzu-Chiao Lin.

**Writing – review & editing:** Tzu-Chiao Lin, Wu-Chien Chien, Je-Ming Hu, Chao-Yang Chen.

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
