## [Decision Letter · Decision Letter 0]

3 Oct 2019

PONE-D-19-23287

Risk of Colorectal Cancer in Patients with Alcoholism: A Nationwide, Population-Based Nested Case-Control Study

PLOS ONE

Dear MD Chen,

Thank you for submitting your manuscript to PLOS ONE. After careful consideration, we feel that it has merit but does not fully meet PLOS ONE’s publication criteria as it currently stands. Therefore, we invite you to submit a revised version of the manuscript that addresses the points raised during the review process.

Both reviewers raised some major concerns that require to be addressed. The authors should effectively respond to these comments in the revised manuscript

We would appreciate receiving your revised manuscript by Nov 02 2019 11:59PM. To enhance the reproducibility of your results, we recommend that if applicable you deposit your laboratory protocols in protocols.io, where a protocol can be assigned its own identifier (DOI) such that it can be cited independently in the future. For instructions see: http://journals.plos.org/plosone/s/submission-guidelines#loc-laboratory-protocols

We look forward to receiving your revised manuscript.

Kind regards,

Yu Ru Kou, PhD

Academic Editor

PLOS ONE

Journal Requirements:

2. In your ethics statement in the manuscript and in the online submission form, please provide additional information about the patient records used in your retrospective study. Specifically, please ensure that you have discussed whether all data were fully anonymized before you accessed them and/or whether the IRB or ethics committee waived the requirement for informed consent. If patients provided informed written consent to have data from their medical records used in research, please include this information.

3. Please include your tables as part of your main manuscript and remove the individual files. Please note that supplementary tables (should remain/ be uploaded) as separate "supporting information" files

Reviewers' comments:

Reviewer's Responses to Questions

**Comments to the Author**

1. Is the manuscript technically sound, and do the data support the conclusions?

Reviewer #1: Yes

Reviewer #2: Yes

2. Has the statistical analysis been performed appropriately and rigorously? 

Reviewer #1: Yes

Reviewer #2: Yes

3. Have the authors made all data underlying the findings in their manuscript fully available?

Reviewer #1: Yes

Reviewer #2: Yes

4. Is the manuscript presented in an intelligible fashion and written in standard English?

Reviewer #1: Yes

Reviewer #2: Yes

5. Review Comments to the Author

Reviewer #1: 1. This is a large-scale population based study, analyzing by national health insurance database.

2. Colorectal cancer is a multifactorial disease. Many chronic systemic diseases and life habit related factors are related to.

3. Using healthy insurance database as an study cohort may reflect a large scale population result, but also confounded by many bias.

4. How about the relationship between alcoholism and treatment result or impact to survival?

5. How about the cost issue? Treatment for CRC patients with alcoholism cost more?

Reviewer #2: In this study, Tzu-Chiao Lin and colleagues analyzed the risk of colorectal cancer in alcoholism by using Taiwan’s National Health Insurance Research Database which was collected data from 2000 to 2013. The results showed significantly higher risk of colorectal cancer in those patients. Furthermore, there was a time-dependent relationship between alcoholism duration and risk of colorectal cancer which strengthen the evidence. Generally, this article is well prepared and few suggest are listed below:

Abstract:

1. “In this study, we aimed to evaluate the association between alcoholism and CRC.” It might be better to use “risk of CRC.”

2. The abbreviation of “OR” was not defined.

Introduction:

3. As mention in list 1, it would be better to directly use “risk of CRC” rather than “CRC” while the aim was proposed.

Method: no suggestion

Results:

4. The authors analyzed variables including DM, HTN, and peptic ulcer. How about the risk of colon polyps and alcoholism? Strongly recommend to add the information in the main results just as the analyses done for investigating the risk of CRC and alcoholism.

Discussion

5. After the suggestion of list 4 was provided, brief discuss is necessary.

6. PLOS authors have the option to publish the peer review history of their article (what does this mean?). If published, this will include your full peer review and any attached files.

Reviewer #1: No

Reviewer #2: Yes: Wen-Wei, Sung

---

## [Author Response · Author response to Decision Letter 0]

11 Nov 2019

Dear editor Yu Ru Kou:

We have revised our manuscript and thoughtfully addressed the critiques provided by reviewers. We have highlighted that the results of this study need to be interpreted wthin the context of some limitations. In addition, we have taken into considerations of suggestions by the reviewers. Accordingly, we have presented renewed study results in our revised manuscript. We have highlighted amendments in red fonts. We are enthusiastic on resubmission of our revised manuscript to your esteemed Journal for publication.

Sincerely yours,

Chao-Yang Chen, MD

Division of Colorectal Surgery, Department of Surgery

Tri-Service General Hospital, National Defense Medical Center

Taipei, Taiwan, Republic of China

---

## [Decision Letter · Decision Letter 1]

30 Jan 2020

PONE-D-19-23287R1

Risk of Colorectal Cancer in Patients with Alcoholism: A Nationwide, Population-Based Nested Case-Control Study

PLOS ONE

Dear MD Chen,

Thank you for submitting your manuscript to PLOS ONE. After careful consideration, we feel that it has merit but does not fully meet PLOS ONE’s publication criteria as it currently stands. Therefore, we invite you to submit a revised version of the manuscript that addresses the points raised during the review process.

Reviwer 3 raised several concerns that need to be adequately addressed or revised.

We would appreciate receiving your revised manuscript by Mar 15 2020 11:59PM. To enhance the reproducibility of your results, we recommend that if applicable you deposit your laboratory protocols in protocols.io, where a protocol can be assigned its own identifier (DOI) such that it can be cited independently in the future. For instructions see: http://journals.plos.org/plosone/s/submission-guidelines#loc-laboratory-protocols

We look forward to receiving your revised manuscript.

Kind regards,

Yu Ru Kou, PhD

Academic Editor

PLOS ONE

Reviewers' comments:

Reviewer's Responses to Questions

**Comments to the Author**

1. If the authors have adequately addressed your comments raised in a previous round of review and you feel that this manuscript is now acceptable for publication, you may indicate that here to bypass the “Comments to the Author” section, enter your conflict of interest statement in the “Confidential to Editor” section, and submit your "Accept" recommendation.

Reviewer #1: All comments have been addressed

Reviewer #2: All comments have been addressed

Reviewer #3: (No Response)

2. Is the manuscript technically sound, and do the data support the conclusions?

Reviewer #1: Partly

Reviewer #2: Yes

Reviewer #3: Partly

3. Has the statistical analysis been performed appropriately and rigorously? 

Reviewer #1: Yes

Reviewer #2: Yes

Reviewer #3: No

4. Have the authors made all data underlying the findings in their manuscript fully available?

Reviewer #1: Yes

Reviewer #2: Yes

Reviewer #3: Yes

5. Is the manuscript presented in an intelligible fashion and written in standard English?

Reviewer #1: No

Reviewer #2: Yes

Reviewer #3: Yes

6. Review Comments to the Author

Reviewer #1: The English writing could be better.

This study is a retrospective cohort study, many limitation of data. However, it did provide some interesting information.

Reviewer #2: The authors answered all questions accordingly. I have no question for this manuscript. It is acceptable for publication.

Reviewer #3: The authors present results from a large-scale populations based study in Taiwan of alcoholism as a risk factor for colorectal cancer (CRC). They find that alcoholism is a risk factor and that there is a time-dependent relationship between the duration of alcoholism and CRC. The manuscript will be strengthened if the authors consider the following points.

1. Controls are matched to cases based on age, gender, index date, and number of medical visits. However, on page 9, authors also mention something about matching cases and controls on residential urbanization level (6th line). I do not think this was actually done based on results presented in Table 1. Authors should clarify their statements.

2. Medical events (including the diagnosis of CRC) were identified from 2000-2013. Since alcoholism and duration of alcoholism are key predictors for the models of interest, were the authors able to go back to 1995 to evaluate this exposure? Authors should clarify the time period for identifying case/control status and the time period reviewable for alcoholism.

3. I am very confused by some of the results presented in Table 2 for the crude OR, in particular, but might carry over to the adjusted OR as well. Specifically, for age group, cases and controls were matched on age and based on Table 1, the percentages in the different age groups between cases and controls are identical. I do not see how the authors find a significant OR for age. I actually used the numbers presented in Table 1 for age group and cases and controls to run a logistic regression and find a non-significant result (p-value of 1). The crude OR for gender (though not significant, as expected, since this was a matching variable) doesn't seem to match what you would get with the numbers presented in Table 1. Similar for other crude ORs presented in Table 2. Authors should carefully check all ORs and confidence intervals and, if needed, clarify what is meant by crude OR (if it is not a simple univariate logistic regression with the listed variable as the single predictor of case status).

Minor points.

1. In the Statistical Analysis section, the authors have a sentence about the dose-response effect of CRC (Further analysis was performed....). This sentence either refers to analyses not presented in the results, or the authors switch CRC and alcoholism. I believe the authors mean to say a dose-response effect of alcoholism on risk of CRC. They further state based on the "average number of medical visits for CRC", but the presented results are based on duration in years of alcoholism, which even if CRC is replaced by alcoholism, doesn't necessarily correspond to number of medical visits. Authors should clarify this statement.

2. on page 11, 2nd paragraph, sentence starting with "In addition, patients in the case group..", "were significantly higher in the case group" should be changed to "than the control group".

3. The note under Table 2 does not list all of the variables included in the adjusted model, since not all variables included in the model are included in the table.

4. Table 4: authors should include information about how many cases/controls had alcoholism by the different durations. Also, in these models was the reference group no alcoholism, so that, for example, in the >5 years results, those with alcoholism for less than 5 years were excluded from the analysis? Also, the note about retrospective duration is confusing, so should be clarified.

5. page 13, 3rd line "is the first large population analysis shows" should be "is the first large population-based analysis that shows"

6. Authors provide supplemental tables but do not refer to them in the text anywhere.

7. PLOS authors have the option to publish the peer review history of their article (what does this mean?). If published, this will include your full peer review and any attached files.

Reviewer #1: No

Reviewer #2: Yes: Wen-Wei Sung

Reviewer #3: No

---

## [Author Response · Author response to Decision Letter 1]

9 Feb 2020

Dear editors and reviewers:

We have revised our manuscript and thoughtfully addressed the critiques provided by reviewers. In addition, we have taken into considerations of suggestions by the reviewers. Accordingly, we have presented renewed study results in our revised manuscript.

---

## [Decision Letter · Decision Letter 2]

11 Mar 2020

PONE-D-19-23287R2

Risk of Colorectal Cancer in Patients with Alcoholism: A Nationwide, Population-Based Nested Case-Control Study

PLOS ONE

Dear MD Chen,

Thank you for submitting your manuscript to PLOS ONE. After careful consideration, we feel that it has merit but does not fully meet PLOS ONE’s publication criteria as it currently stands. Therefore, we invite you to submit a revised version of the manuscript that addresses the points raised during the review process.

The reviewer still had some minor comments that need to be addressed.

We would appreciate receiving your revised manuscript by Apr 25 2020 11:59PM. To enhance the reproducibility of your results, we recommend that if applicable you deposit your laboratory protocols in protocols.io, where a protocol can be assigned its own identifier (DOI) such that it can be cited independently in the future. For instructions see: http://journals.plos.org/plosone/s/submission-guidelines#loc-laboratory-protocols

We look forward to receiving your revised manuscript.

Kind regards,

Yu Ru Kou, PhD

Academic Editor

PLOS ONE

Journal Requirements:

Additional Editor Comments (if provided):

Reviewers' comments:

Reviewer's Responses to Questions

**Comments to the Author**

1. If the authors have adequately addressed your comments raised in a previous round of review and you feel that this manuscript is now acceptable for publication, you may indicate that here to bypass the “Comments to the Author” section, enter your conflict of interest statement in the “Confidential to Editor” section, and submit your "Accept" recommendation.

Reviewer #3: (No Response)

2. Is the manuscript technically sound, and do the data support the conclusions?

Reviewer #3: Yes

3. Has the statistical analysis been performed appropriately and rigorously? 

Reviewer #3: Yes

4. Have the authors made all data underlying the findings in their manuscript fully available?

Reviewer #3: Yes

5. Is the manuscript presented in an intelligible fashion and written in standard English?

Reviewer #3: Yes

6. Review Comments to the Author

Reviewer #3: The authors have addressed the majority of my earlier comments. There remain a few minor points that should be addressed:

1. On page 11, the authors added a sentence "In addition...". The previous statement already states that there are differences between groups in comorbidities, so the authors may want to use "In particular" instead. They also end the sentence after "comorbidities" but that makes the next sentence an incomplete sentence.

2. On page 11, the authors state that most CRC patients were treated in Northern and Southern Taiwan - according to Table 1, Middle Taiwan should also be stated, since a slightly higher percentage of CRC patients came from there than from Southern Taiwan.

3. On page 11, "an combination" should be "a combination"

4. On page 11, last paragraph, the authors mention that Table 2 includes univariate results. The authors opted to remove these results from the table based on my earlier critique (I actually think they are useful, though the authors would then need to check results, because their explanation in their response for the age difference that I questioned is not adequate). I leave it to the editor to decide whether the univariate results need to be presented. If they are not, authors should not refer to them in the text.

5. On page 12, the authors mention findings from stratified analyses based on urbanization levels and healthcare levels without giving relevant statistics or data. Authors are encouraged to either include OR and 95% CI in the text to support the statement, add the results to Table 3 or add a supplemental table that includes the information.

7. PLOS authors have the option to publish the peer review history of their article (what does this mean?). If published, this will include your full peer review and any attached files.

Reviewer #3: No

---

## [Author Response · Author response to Decision Letter 2]

15 Mar 2020

Dear editors and reviewers:

We have revised our manuscript and thoughtfully addressed the critiques provided by reviewers. In addition, we have taken into considerations of suggestions by the reviewers. We have highlighted amendments in red fonts.

---

## [Decision Letter · Decision Letter 3]

22 Apr 2020

Risk of Colorectal Cancer in Patients with Alcoholism: A Nationwide, Population-Based Nested Case-Control Study

PONE-D-19-23287R3

Dear Dr. Chen,

We are pleased to inform you that your manuscript has been judged scientifically suitable for publication and will be formally accepted for publication once it complies with all outstanding technical requirements.

With kind regards,

Yu Ru Kou, PhD

Academic Editor

PLOS ONE

Additional Editor Comments (optional):

Reviewers' comments:

Reviewer's Responses to Questions

**Comments to the Author**

1. If the authors have adequately addressed your comments raised in a previous round of review and you feel that this manuscript is now acceptable for publication, you may indicate that here to bypass the “Comments to the Author” section, enter your conflict of interest statement in the “Confidential to Editor” section, and submit your "Accept" recommendation.

Reviewer #3: All comments have been addressed

2. Is the manuscript technically sound, and do the data support the conclusions?

Reviewer #3: (No Response)

3. Has the statistical analysis been performed appropriately and rigorously? 

Reviewer #3: (No Response)

4. Have the authors made all data underlying the findings in their manuscript fully available?

Reviewer #3: (No Response)

5. Is the manuscript presented in an intelligible fashion and written in standard English?

Reviewer #3: (No Response)

6. Review Comments to the Author

Reviewer #3: (No Response)

7. PLOS authors have the option to publish the peer review history of their article (what does this mean?). If published, this will include your full peer review and any attached files.

Reviewer #3: No

---

## [Editor Report · Acceptance letter]

1 May 2020

PONE-D-19-23287R3 

Risk of Colorectal Cancer in Patients with Alcoholism: A Nationwide, Population-Based Nested Case-Control Study 

Dear Dr. Chen:

I am pleased to inform you that your manuscript has been deemed suitable for publication in PLOS ONE. Congratulations! Your manuscript is now with our production department. 

With kind regards,

on behalf of

Dr. Yu Ru Kou 

Academic Editor

PLOS ONE